# Learning to Remember, Learn, and Forget in Attention-Based Models

## Abstract

The ability to perform learning during inference (Brown et al., 2020a), *i.e.* in-context learning (ICL) is a core feature of self-attention in transformers. ICL acts like an online associative memory and is believed to underpin transformers' capabilities in complex sequence processing tasks. In some cases, ICL was shown to simulate online gradient descent of a local loss function on an input sequence. In this work, we view ICL as a continual learning problem that may suffer from memory interference and requires a solution to a plasticity–stability dilemma. We examine here the memory consolidation properties of ICL and propose a Bayesian continual learning framework to solve this dilemma, leading to a new attention model. Our framework builds on the idea of metaplasticity in neuroscience, where the level of plasticity of each synapse is tied to an importance measure grounded by a Bayesian prior distribution capturing previously learned knowledge. Our approach explains several gated linear attention models in the literature, identifying the respective assumptions from a Bayesian learning perspective. Furthermore, our Bayesian continual learning approach provides a principled approach to forgetting, enabling the design of attention layers with a desired memory horizon. Our experiments achieve competitive performances on synthetic benchmarks. Additionally, we experiment on several commonsense reasoning benchmarks where small models benefit from consolidated synapses, outperforming strong baseline like Gated Delta Networks.

## 1 Introduction

Transformers have been established as the workhorse of generative AI, underpinning breakthroughs in large language modeling (Brown et al., 2020b; Chowdhery et al., 2023), computer vision (Dosovitskiy et al., 2020), robotics (Brohan et al., 2022), and even scientific domains like chemistry (Schwaller et al., 2019), and biology (Jumper et al., 2021). Central to this success is the *self-attention* mechanism (Vaswani et al., 2017), which allows a model to capture interactions among all tokens in a sequence in a highly parallelizable manner. However, this mechanism requires caching key-value (KV) pairs for each input token, causing memory and computational costs to grow linearly and quadratically, respectively, with sequence length. Consequently, long-context data scenarios pose strong technical challenges for classical Transformer architectures, especially at the edge (Kim et al., 2023).

A promising direction to simultaneously address the computational and memory bottlenecks is to adopt *fixed-size* attention memories, such as in linear transformers (Schlag et al., 2021) and state space models (Gu and Dao, 2023a). These trade the dynamically growing KV cache with a fixed-size memory state that is updated at each token. A key insight is their ability to perform *in-context learning (ICL)* (Brown et al., 2020a), whereby the model effectively "learns" at test time by processing contextual examples within the input. ICL solves a continual and online learning problem, not unlike how the brain dynamically adjusts its weights through neural and synaptic plasticity. In this view, we know that learning with fixed-size memory without revisiting past states and inputs (*i.e.* replay) can lead to catastrophic interference (McCloskey and Cohen, 1989). This interference likely contributes to the limited capacity of linear transformers (Schlag et al., 2021).

The concept of *metaplasticity* from neuroscience (Abraham and Bear, 1996), which posits that the degree of plasticity is itself adaptive to preserve prior knowledge, has led to practical algorithms to

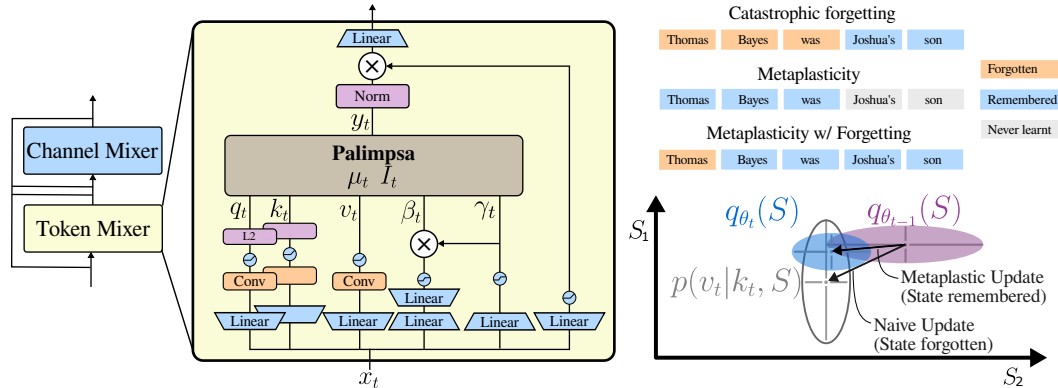

Figure 1: Bayesian Metaplasticity Attention. (Left) Palimpsa is a novel token mixer (attention layer) that uses Bayesian inference at test time to compute attention scores. (Right) Self-attention in autoregressive transformers is inherently a continual learning problem, and as such can suffer from catastrophic forgetting. Metaplasticity is a widely studied technique that dynamically modifies the learning rate to prevent important prior information to be forgotten. In this work, we derive Bayesian Metaplastic Attention, a new attention block based on an online Bayesian posterior, preventing both catastrophic forgetting and catastrophic remembering. (Bottom Right) $q_{\theta_t}$ is the (variational) distribution over memory states $S$ at time step $t$ tracked in Palimpsa. The Bayesian metaplastic approach provably updates the distribution in a way that retains information essential to the task (purple to blue region). Whereas a Naive update would discard prior information in a detrimental way. Palimpsa is further equipped with a forgetting mechanism to present loss of plasticity at long sequences, by discarding stale information.

solve the catastrophic forgetting problem (Zenke et al., 2017a; Kirkpatrick et al., 2017a; Benna and Fusi, 2016). In this work, we pose ICL as a continual learning problem that can be solved through metaplasticity, building on prior work that associated ICL and gradient descent (Akyürek et al., 2022; von Oswald et al., 2023a). However, existing metaplasticity methods are unsuitable for transformers for the following reasons: They either rely on clearly demarcated tasks and associated labels, growing memory, and lack differentiability (Kudithipudi et al., 2023). As we aim to introduce metaplastic methods embedded in transformer architectures, differentiability is necessary for training. One natural solution to these challenges is to formulate memory updates as a process of *Bayesian Gradient Descent (BGD)*, ensuring that each update balances prior knowledge with new evidence (Zeno et al., 2021). These methods effectively adjust parameters according to their uncertainty in an online fashion. However, BGD and other metaplasticity methods suffer from catastrophic *remembering* (Kaushik et al., 2021), induced by a loss of plasticity that *suppresses* the ability to learn new knowledge to preserve old ones. For ICL, this would imply that the model becomes unable to incorporate new information past a critical sequence length, which is often cited as a key limitation of state space models (Behrouz et al., 2024). The recent Metaplasticity from Synaptic Uncertainty (MESU) solves this problem by leveraging a Bayesian framework to also enable the forgetting by discarding stale or unused information (Bonnet et al., 2025a). The degree of this *palimpsest* property [1] can be adjusted through a Bayesian prior, which effectively controls the time horizon of the memory.

Building on MESU, we introduce *Palimpsa*, a dual-state attention block that performs metaplastic Bayesian updates to a fixed-size attention memory. Palimpsa mitigates in-context catastrophic forgetting by adjusting per-state update magnitude based on memory uncertainty, preserving critical past information. By releasing information predicted as stale, it also prevents in-context catastrophic remembering. The key to deriving Palimpsa is the formulation of self-attention as an optimization of an inner variational free energy at test-time, for which the variational posterior updates can be analytically computed. Despite being based on Bayesian learning, our formulation scales efficiently on GPUs at speeds comparable to Mamba and other gated state-space models (Gu and Dao, 2024; Yang et al., 2024). We designed custom Triton kernels to accelerate key operations in the attention

---

[1]A palimpsest is a writing surface where the original text is scraped or washed off to be reused, especially used in ancient works when parchment was of limited supply – similarly to the fixed-size memory in Palimpsa

block, which are made publicly available[2].

Furthermore, we find that existing related methods can be derived from specific *assumptions* on the variational posterior, connecting several prior work in a common mathematical framework. We validate Palimpsa on synthetic benchmarks and commonsense reasoning language benchmarks. Our results demonstrate competive performance across different memory sizes especially for smaller scale models, highlighting robust performance under tight memory constraints.

**Our specific contributions are:**

- Introduce the neuroscience concept of metaplasticity to mitigate in-context catastrophic forgetting in transformer models.
- Palimpsa, a model that prevents in-context catastrophic rememberingby gradually releasing outdated knowledge, with scalable custom kernels.
- A mathematical framework that casts several state-space models as special cases of a common Bayesian framework

RELATED WORK

**Continual Learning Methods and Neural Sequence Models**   Lifelong learning or Continual Learning (CL) models aims to learn new tasks sequentially without forgetting previously learned tasks (Kudithipudi et al., 2022). CL is a critical ability for adapting to dynamic real-world environments. However, these models face a limitation in mitigating catastrophic forgetting when adapting to new data, making it challenging to achieve a balance between learning new knowledge and retaining prior knowledge, also known as *stability-plasticity dilemma* (Fusi et al., 2005a). CL methods can be broadly categorized into *replay-based* ((Buzzega et al., 2020), (Lopez-Paz and Ranzato, 2017)), *dynamic network expansion* ((Rusu et al., 2016), (Wang et al., 2024)), and *regularization-based* methods ((Zenke et al., 2017b), (Kirkpatrick et al., 2017b)). While these approaches are complementary mechanisms for CL systems (McClelland et al., 1995), our work focuses specifically on regularization-based metaplasticity, because it provides a local and efficient to solve the stability-plasticity dilemma using finite size memory.

CL has been considered in State Space Models (SSM), but differently to Palimpsa. (Cheng et al., 2024) used the subspace projection methods in the token mixer, but did not study the ICL aspects of this problem. Other work applied modern transformer and SSM models to improve on CL benchmarks (Thengane et al., 2022). Here, we instead show how these mechanisms can improve general language modeling tasks that do not have an explicitly constructed CL structure, namely language.

Another related concept is Bayesian learning in terms of ICL. (Xie et al., 2021) analyzed ICL within the Bayesian framework, assuming access to the nominal language distribution and that the tokens are generated from a hidden Markov model. (Wies et al., 2023) relaxed the model assumption and assumed access to a pretrained model that is close to the nominal distribution conditioned on any token sequence. Other work (Arora et al., 2024; Hahn and Goyal, 2023; Zhang et al., 2023; Falck et al., 2024) provided Bayesian analysis of ICL, but did not address the learning dynamics of ICL with Bayesian learning.

Bayesian learning and metaplasticity are closely linked in the hypothesis that synapses maintain uncertainty estimates (Aitchison et al., 2021) to dynamically adjust their plasticity. This perspective inspired metaplastic continual learning methods like MESU (Bonnet et al., 2025a). This type of metaplasticity can be written in a local and linear fashion. In Palimpsa, we take advantage of these properties to allow scalable training on GPUs using the techniques employed for gated state-space models (Gu and Dao, 2023a).

**Gated Linear Recurrence Models**   Linear attention mechanisms for autoregressive language modeling were first suggested by (Katharopoulos et al., 2020) due to their promising compute and memory complexity advantage over self-attention. (Schlag et al., 2021) demonstrated the equivalence of linear self-attention and fast weight programming and highlighted their limited ability to perform recall tasks. The introduction of data-dependent gating and forgetting mechanisms was shown to mitigate the limited capacity of linear attention (Yang et al., 2024; 2023; Liu et al., 2024a; Sun et al.,

---

[2]Link omitted for double-blind review

2024a; Beck et al., 2024), along with efficient GPU I/O-aware implementations enabling scaling for large language modeling.

(Akyürek et al., 2022; von Oswald et al., 2023b) demonstrated that the forward pass through the linear attention mechanism can be framed as an iterative gradient descent on a regression cost function, thus enabling test-time adaptation of the model underpinning ICL. With this mechanistic perspective on ICL, the choice of the model and the cost function directly influenced the resulting gating and forgetting mechanisms (Yang et al., 2023). The perspective of gradient descent on local loss functions was further elaborated in Mesa-optimization (von Oswald et al., 2023a; 2025), Test-Time Training (Sun et al., 2023), Gated Delta Networks (Yang et al., 2024), Longhorn (Liu et al., 2024a) and Titans (Behrouz et al., 2024). For example, Longhorn (Liu et al., 2024a) applied an online learning framework with to this relationship to derive gating mechanisms from first principles.

While metaplasticity was never linked with the self-attention mechanisms or gated state-space models, the gating mechanisms could be seen as a form of suboptimal metaplasticity. Furthermore, while gating enables the model to control which information is compressed into the hidden state achieving a form of metaplasticity, there was no efficient mechanism that enables the controlled eviction of stale information was proposed.

Palimpsa is closest to Longhorn (Liu et al., 2024a) and Mesanet (von Oswald et al., 2025). The Longhorn update is derived from a stability-plasticity loss reminiscent of metaplasticity. There, a trainable projection of the input tokens $\beta_t$ at each time step determines the stability-plasticity ratio for each row of the fixed-size memory matrix. When $\beta_{t_i} = 0$, the entire $i$-th row of the memory will remain unchanged. Furthermore, within a sequence, every longhorn memory state will have the same plasticity rate, *i.e.* no in-context metaplasticity.

In contrast, our Bayesian metaplasticity perspective dictates a plasticity rate that changes *in-context*. This is achieved using a second state, as in the Mesanet. Mesanet (von Oswald et al., 2025) seeks the optimal solution of an in-context regression objective, leading to an update that is strikingly similar with Palimpsa. This is because Bayesian and frequentist solutions to linear regression problems lead to identical estimates of the mean. However, the Bayesian approach also offers a per-state uncertainty measure, and a perspective that leads to assumptions that are different from those of Mesanet. As described in the methods, metaplasticity on each synapse dictates the objective to have a vector $\beta_t$, weighting the contribution to the loss function on each component (as in Longhorn), and leading to a synaptic *importance matrix*, defined as the inverse of the synaptic variance. To maintain training throughput, preserving the principle of metaplasticity entails a diagonal approximation of the covariance matrix. Mesanet incorporates such correlations with a scalable model, computing iteratively by gradient descent the inverse of the covariance matrix in an highly parallelised manner. This entails a numerical approximation, costing inner loops updates and a *scalar* $\beta_t$. As such, in Mesanet, every line of the memory matrix has the same covariance matrix, meaning that importance is shared along a an entire dimension. Maintaining the per-state importance as in Palimpsa is beneficial to control stability–plasticity trade-off. Furthermore, Palimpsa ties input integration to forgetting by design (see Methods). Finally, because Palimpsa is Bayesian, the attention head provides output uncertainty that can be leveraged to improve overall model performance.

## 2 BACKGROUND AND METHODS

### 2.1 BACKGROUND: SELF-ATTENTION HEADS AND METAPLASTIC MEMORY

The decoder self-attention (Vaswani et al., 2017) autoregressively maps the sequence $\{\boldsymbol{x}_t\}_{t=1}^L$ into $\{\boldsymbol{y}_t\}_{t=1}^L$. Each token $\boldsymbol{x}_t$ is projected into key, query, and value vectors $\boldsymbol{k}_t, \boldsymbol{q}_t \in \mathbb{R}^{d_k}, \boldsymbol{v}_t \in \mathbb{R}^{d_v}$. Then, self-attention computes a weighted combination of the values $\boldsymbol{V}_t = [\boldsymbol{V}_{t-1}, \boldsymbol{v}_t]$ based on the similarity between each query $\boldsymbol{q}_t$ and the keys $\boldsymbol{K}_t = [\boldsymbol{K}_{t-1}, \boldsymbol{k}_t]$. Concretely, this can be expressed as:

$$\boldsymbol{y}_t = \boldsymbol{V}_t \operatorname{Softmax}\big(\boldsymbol{K}_t^\top \boldsymbol{q}_t\big).$$

The similarity between queries and keys (the "look-up" step) determines the extent to which each value contributes to the output. Linear Transformers (Schlag et al., 2021) omit the softmax, in which case the KV cache can be replaced by a fixed-size matrix, updated in-place for every new incoming token with a Hebbian learning update

$$\boldsymbol{S}_t = \boldsymbol{S}_{t-1} + \boldsymbol{v}_t \otimes \boldsymbol{k}_t, \quad \text{and} \quad \boldsymbol{y}_t = \boldsymbol{S}_t \boldsymbol{q}_t, \text{ where } \boldsymbol{S}_t \in \mathbb{R}^{d_v \times d_k},$$

which we refer to as the *attention memory*, collects all past key–value information on-the-fly and leading to constant memory usage and linear compute in sequence length $L$.

However, if the new key $\boldsymbol{k}_t$ is not orthogonal to previously stored keys, the weight update will partially overwrite older memories. (Schlag et al., 2021) demonstrated the limited capacity of linear transformers and implemented an error-correcting delta rule to mitigate this issue. This idea has been later expanded over to improve performance and trainability (Sun et al., 2024b; Liu et al., 2024a; Yang et al., 2024).

However, a dilemma emerges with long sequences: should the model prioritize learning to map new key–value pairs, or preserve the relationships from earlier pairs? Continual learning assumes that some parameters are important (e.g. because they are shared across tasks or critical to one) and should be preserved, while others are expendable. The learning rate for each weight can be adjusted based on *importance* to address this: More important weights adopt a smaller learning rate to maintain their established mappings, while less important ones can adapt more rapidly. Such dependence of plasticity on past cues is known as *metaplasticity* in neuroscience (Fusi et al., 2005b).

Recently, (Bonnet et al., 2025a) proposed a mathematically grounded approach to derive a metaplastic learning rule from a Bayesian framework. The key distinguishing element there is to *also* formalize forgetting in an online life-long learning scenario, thereby preventing a vanishing learning rate for every weight. It thus avoids the so called catastrophic remembering (Kaushik et al., 2021) problem in traditional Bayesian frameworks. Building on this idea, we treat the self-attention memory update as a Bayesian inference problem that includes formalized forgetting. Our self-attention mechanism is responsible for remembering, learning, and forgetting the context, which is why the training of Palimpsa can be understood as *Learning to Remember, Learn, and Forget*.

## 2.2 DERIVATION OF PALIMPSA

**Bayesian Formulation of the Attention Objective** For a given time step, the attention head can be defined as $p(\boldsymbol{v}|\boldsymbol{k}, \boldsymbol{\beta}, \boldsymbol{S})$. Given the inputs $\boldsymbol{k} \in \mathbb{R}^{d_k}$ and $\boldsymbol{\beta} \in \mathbb{R}^{d_v}$, the attention head assigns a probability to each value $\boldsymbol{v} \in \mathbb{R}^{d_v}$ using its state $\boldsymbol{S} \in \mathbb{R}^{d_v \times d_k}$. Assuming a Gaussian distribution, we define the objective function (Linear regression) of the attention head as the negative-log-likelihood:

$$-\log p(\boldsymbol{v} \mid \boldsymbol{k}, \boldsymbol{\beta}, \boldsymbol{S}) = \tfrac{1}{2}\|\boldsymbol{S}\,\boldsymbol{k} - \boldsymbol{v}\|^2_{\mathrm{diag}(\boldsymbol{\beta})}.$$

Given $t$ tokens $\{\boldsymbol{x}_j\}_{j=1}^t$ with $\boldsymbol{x}_j \in \mathbb{R}^{d \times 1}$, we obtain $t$ training data points $\{\boldsymbol{d}_j\}_{j=1}^t = \{(\boldsymbol{k}_j, \boldsymbol{\beta}_j), \boldsymbol{v}_j\}_{j=1}^t$. Applying Bayes' rule to these observations gives the posterior distribution on $\boldsymbol{S}$, which can be expressed in the direct or recursive form, respectively:

$$p(\boldsymbol{S}|\boldsymbol{d}_{1:t}) = \frac{p(\boldsymbol{d}_{1:t}|\boldsymbol{S}) \cdot p(\boldsymbol{S})}{p(\boldsymbol{d}_{1:t})} = \frac{p(\boldsymbol{d}_t|\boldsymbol{S}) \cdot p(\boldsymbol{S}|\boldsymbol{d}_{1:t-1})}{p(\boldsymbol{d}_t)}. \tag{1}$$

Since the query $\boldsymbol{q}_t$ is independent of $\boldsymbol{S}$, the output $\boldsymbol{y}_t$ is the expected value recalled by $\boldsymbol{q}_t$:

$$\boldsymbol{y}_t = \mathbb{E}_{p(\boldsymbol{S}|\boldsymbol{d}_{1:t})}\left[\boldsymbol{S}\boldsymbol{q}_t\right] = \mathbb{E}_{p(\boldsymbol{S}|\boldsymbol{d}_{1:t})}\left[\boldsymbol{S}\right]\boldsymbol{q}_t = \boldsymbol{\mu}_t\boldsymbol{q}_t. \tag{2}$$

In the case of an unbounded context size, catastrophic remembering (Kaushik et al., 2021) may occur: As $t$ increases, the prior becomes too dominant, and no additional information can be integrated from new tokens. Bayesian forgetting can be introduced by *recursively* computed by truncating the posterior $p(\boldsymbol{S}|\boldsymbol{d}_{t-N:t})$, where $N$ is the size of the memory window (Bonnet et al., 2025a). Inspired by common practice in state space models (Gu and Dao, 2023b), we introduce input-dependence to the Bayesian forgetting (see Appendices).
This suggests a simple interpretation: at each time step, we discard $\frac{\gamma_t}{N}$ from all previous data we have seen, $\gamma_t \in [0, 1]$. Effectively our weighted posterior never has more weight than that of a truncated posterior with a memory window of size $N$. Thus, with a context window $N$ that reflects the maximum memory capacity of the model, catastrophic remembering can be avoided. We further tie the forgetting and the input gates by defining: $\gamma_t = \sigma(\boldsymbol{\theta}_\gamma \boldsymbol{x}_t)$, and $\boldsymbol{\beta}_t = \gamma_t \sigma(\boldsymbol{\theta}_\beta \boldsymbol{x}_t)$. In this way, no forgetting simultaneously entails no input.

**Framing Bayesian Attention as an Optimization Problem** We use variational inference to recast Bayesian inference as an optimization problem (Blundell et al., 2015). For each time step $t$, We define $q_{\boldsymbol{\theta}_{t,i}}(\boldsymbol{S}_i)$, $i = 1, \ldots, d_v$, a variational distribution over $\boldsymbol{S}_i$ parameterized by $\boldsymbol{\theta}_{t,i}$. We minimize the

Kullback–Leibler (KL) divergence between the variational distribution and the true posterior by finding the optimal $\boldsymbol{\theta}_{t,i}$:

$$\boldsymbol{\theta}_{t,i} = \arg\min_{\boldsymbol{\theta}} D_{\text{KL}}\left[q_{\boldsymbol{\theta}}(\boldsymbol{S}_i) \,\|\, p(\boldsymbol{S}_i|\boldsymbol{d}_{1:t})\right], \; i = 1, \ldots, d_v.$$

The variational distribution $q_{\boldsymbol{\theta}_{t,i}}$ is modeled as a multivariate Gaussian of dimension $d_k$:

$$q_{\boldsymbol{\theta}_{t,i}}(\boldsymbol{S}) \sim \mathcal{N}(\boldsymbol{\mu}_{t,i}, \boldsymbol{\Sigma}_{t,i}), \text{where } \boldsymbol{\theta}_{t,i} = \{\boldsymbol{\mu}_{t,i}, \boldsymbol{\Sigma}_{t,i}\}, \, \boldsymbol{\mu}_{t,i} \in \mathbb{R}^{d_k}, \, \boldsymbol{\Sigma}_{t,i} \in \mathbb{R}^{d_k \times d_k}, \, i = 1, \ldots, d_v.$$

Using Bayes' theorem and the definition of KL divergence, finding the optimal $\boldsymbol{\theta}_{t,i}$ is equivalent to minimizing the free energy:

$$\mathcal{F}_{t,i} = D_{\text{KL}}\left[q_{\boldsymbol{\theta}_{t,i}}(\boldsymbol{S}_i) \,\|\, p(\boldsymbol{S}_i)\right] - \mathbb{E}_{q_{\boldsymbol{\theta}_{t,i}}(\boldsymbol{S}_i)}\left[\sum_{s=1}^{t} \log p(\boldsymbol{d}_s|\boldsymbol{S}_i)\right].$$

This free variational energy is analytically tractable, and its minimum with respect to $\boldsymbol{\mu}_{t,i}$ and $\boldsymbol{\Sigma}_{t,i}$ can be computed in closed form (see Appendices) without requiring any approximation.

**Palimpsa Layer and Architecture** We minimise the free-energy $\mathcal{F}_{t,i}$ by setting its gradient to zero, resulting in the following exact posterior updates (see Appendices). For computational tractability, Palimpsa only keeps the diagonal term of the covariance matrices ($\boldsymbol{\Sigma}_{t,i} = \text{diag}(\boldsymbol{\sigma}_{t,i}^2)$) as metaplasticity dictates. This results in the following update equation that can be computed chunk-wise with cumulative sum:

$$\boldsymbol{I}_t = \alpha_t \boldsymbol{I}_{t-1} + (\alpha_t - 1)\boldsymbol{I}_{prior} + \boldsymbol{\beta}_t \otimes \boldsymbol{k}_t^2, \quad \boldsymbol{\mu}_t = \alpha_t \frac{\boldsymbol{I}_{t-1}}{\boldsymbol{I}_t} \odot \boldsymbol{\mu}_{t-1} + \frac{1}{\boldsymbol{I}_t} \odot \left[(\boldsymbol{\beta}_t \odot \boldsymbol{v}_t) \otimes \boldsymbol{k}_t\right]. \quad (3)$$

Where $\alpha_t := (1 - \frac{\gamma_t}{N})$, $\boldsymbol{I}_t := \frac{1}{\boldsymbol{\sigma}_t^2}$ a precision matrix representing the importance of each state, and $I_{prior}$ is the importance prior. Our experiments used the Palimpsa layer as a drop-in replacement of the Gated Delta Rule (Yang et al., 2024), with modifications related for $\boldsymbol{\beta}_t$ and $\gamma_t$. We tested in three size configurations, namely 170M, 340M and 760M (see appendices for details). In our implementation, the size of the memory window is trained, as $N := N_{init} \exp(logN)$, where $logN$ is the trained parameter. For language modeling, we choose the state size to be half that of Gated Deltanet ($d_k$ is divided by two compared to Gated DeltaNet) so that the state sizes remain comparable.

Bayesian Inference at Test-Time as a General Framework of Gated Models

We investigate previous works in the light of the Bayesian view of ICL. Writing out the variational free energy above with terms that depend on $\boldsymbol{\mu}_t$ (see Appendices), we can identify three components:

$$\mathcal{F}_{t,i}(\boldsymbol{\mu}_i) = \underbrace{\frac{1}{2}\|\boldsymbol{\mu}_i \boldsymbol{k}_t - v_{t,i}\|_{(\beta_{t,i})}^2}_{\text{plasticity}} + \underbrace{(1 - \alpha_t)\frac{\boldsymbol{\mu}_i^2}{I_{prior}}}_{\text{forgetting}} + \underbrace{\alpha_t(\boldsymbol{\mu}_{t-1,i} - \boldsymbol{\mu}_i)^T \boldsymbol{\Sigma}_{t-1,i}^{-1}(\boldsymbol{\mu}_{t-1,i} - \boldsymbol{\mu}_i)}_{\text{stability}}.$$

The first term is similar to other fixed-term memories, and contributes to adding new knowledge. The second term introduces the learning window, through its dependence on $N$, it determines how far in the past the memories are stored, and then forgotten. The third term is the one that determines the plasticity rate based on the importance of the synapse. A major challenge in minimizing $\mathcal{F}_{t,i}$ is calculating the inverse of the matrix $\boldsymbol{\Sigma}_{t-1,i}^{-1} \in \mathbb{R}^{d_k \times d_k}$. (von Oswald et al., 2023a) use the Sherman-Morrison formula to compute it step-by-step but this approach doesn't scale well. Another approach is to use conjugate gradients, as done in MesaNet (von Oswald et al., 2025). However, both methods require $\beta \in \mathbb{R}$ (a single number) for tractability, since a vector $\boldsymbol{\beta}$ would require the inversion for every row of the state matrix. However, using a single number for $\beta$ is that the stability part of the equation becomes the same for every row.

Longhorn, Deltanets and Palimpsa use at least one approximation for tractability. A common simplification is to assume the matrix $\boldsymbol{\Sigma}_{t-1,i}^{-1}$ is diagonal (meaning it only has values on its main diagonal). Longhorn and Palimpsa use a vector for $\boldsymbol{\beta}$ and thus the diagonal simplification to solve the objective. In contrast, DeltaNets solve their objective by assuming the loss is linear around the current states. In Palimpsa, we further identify this diagonal $\boldsymbol{I}_{t-1,i} \in \mathbb{R}^{d_k}$ with the "importance" of given synapse, and provides its optimal learning rate. For all other gated models in Table 1, the

Table 1: A detailed comparison of layer architectures (see appendices for details). For each layer, this table presents the key design choices and the resulting online update rule. All bold letters denote vectors or matrices. The outer product is written as $\boldsymbol{u}\boldsymbol{v}^\top$ or, where emphasized, as $\boldsymbol{u} \otimes \boldsymbol{v}$. $\mathbb{1}$ is a vector or matrix of ones.

| Layer | Stability Matrix | Input Gating | Forgetting | Objective resolution |
|---|---|---|---|---|
| **Longhorn** | $\mathbb{I}_{d_k}$ | $\boldsymbol{\beta}_t \in \mathbb{R}^{d_v}$ | None | Diagonal approximation |
| | *Update Rule:* $\boldsymbol{\mu}_t = (\mathbb{1}_{d_v \times d_k} - \frac{\boldsymbol{\beta}_t \otimes \boldsymbol{k}_t^2}{1+\boldsymbol{\beta}_t \boldsymbol{k}_t^\top \boldsymbol{k}_t}) \odot \boldsymbol{\mu}_{t-1} + \frac{\boldsymbol{\beta}_t \odot \boldsymbol{v}_t}{1+\boldsymbol{\beta}_t \boldsymbol{k}_t^\top \boldsymbol{k}_t} \otimes \boldsymbol{k}_t$ | | | |
| **DeltaNet** | $\mathbb{I}_{d_k}$ | $\beta_t \in \mathbb{R}$ | None | First order approximation |
| | *Update Rule:* $\boldsymbol{\mu}_t = \boldsymbol{\mu}_{t-1}(\mathbb{I}_{d_k} - \beta_t \boldsymbol{k}_t \boldsymbol{k}_t^\top) + \beta_t \boldsymbol{v}_t \boldsymbol{k}_t^\top$ | | | |
| **Gated DeltaNet** | $\mathbb{I}_{d_k}$ | $\beta_t \in \mathbb{R}$ | $\alpha_t \in \mathbb{R}$ | First order approximation |
| | *Update Rule:* $\boldsymbol{\mu}_t = \boldsymbol{\mu}_{t-1}(\alpha_t(\mathbb{I}_{d_k} - \beta_t \boldsymbol{k}_t \boldsymbol{k}_t^\top)) + \beta_t \boldsymbol{v}_t \boldsymbol{k}_t^\top$ | | | |
| **Mesa** | $\boldsymbol{\Sigma}_t^{-1}$ | $\beta_t \in \mathbb{R}$ | $\alpha_t \in \mathbb{R}$ | Conjugate grad. approx. |
| | *Update Rule:* $\boldsymbol{\Sigma}_t^{-1} = \alpha_t \boldsymbol{\Sigma}_{t-1}^{-1} + (\alpha_t - 1)\boldsymbol{I}_{prior} + \beta_t \boldsymbol{k}_t \boldsymbol{k}_t^\top$ $\boldsymbol{\mu}_t = \boldsymbol{\Sigma}_t\left[\alpha_t \boldsymbol{\Sigma}_{t-1}^{-1}\boldsymbol{\mu}_{t-1} + \beta_t \boldsymbol{v}_t \boldsymbol{k}_t^\top\right]$ | | | |
| **Palimpsa** | $\mathrm{diag}(\boldsymbol{I}_{t-1,i})$ | $\boldsymbol{\beta}_t \in \mathbb{R}^{d_v}$ | $\alpha_t \in \mathbb{R}$ | Diagonal approximation |
| | *Update Rule:* $\boldsymbol{I}_t = \alpha_t \boldsymbol{I}_{t-1} + (\alpha_t - 1)\boldsymbol{I}_{prior} + \boldsymbol{\beta}_t \otimes \boldsymbol{k}_t^2,$ $\boldsymbol{\mu}_t = \alpha_t \frac{\boldsymbol{I}_{t-1}}{\boldsymbol{I}_t} \odot \boldsymbol{\mu}_{t-1} + \frac{1}{\boldsymbol{I}_t} \odot \left[(\boldsymbol{\beta}_t \odot \boldsymbol{v}_t) \otimes \boldsymbol{k}_t\right]$ | | | |
| **Palimpsa** $\left\|\frac{\boldsymbol{I}_t - \boldsymbol{I}_{prior}}{\boldsymbol{I}_{prior}}\right\| \ll 1$ ($\cong$ Mamba2, see text) | *Update Rule:* $\boldsymbol{\mu}_t = \alpha_t \boldsymbol{\mu}_{t-1} + \frac{\boldsymbol{\beta} \odot \boldsymbol{v}_t}{\boldsymbol{I}_{prior}} \otimes \boldsymbol{k}_t$ | | | |

importance is fixed, so $\boldsymbol{I}_{t-1,i} = \vec{1}$. In other words, their stability term is constant, meaning they have no metaplasticity.

Furthermore, we find that Mamba2 is a special case of Palimpsa. In our update equation, for very strong forgetting *i.e.* $I_{prior} \gg \frac{N}{d_k}$, then $\boldsymbol{I}_t \cong \boldsymbol{I}_{prior}$. In this case, $\boldsymbol{I}_t$ would be constant and the update rule simplifies to:

$$\boldsymbol{\mu}_t = \alpha_t \boldsymbol{\mu}_{t-1} + \frac{\boldsymbol{\beta} \odot \boldsymbol{v}_t}{I_{prior}} \otimes \boldsymbol{k}_t,$$

which takes the same form as Mamba2's update rule. While Mamba2 was described as solving a negative inner-product loss (Yang et al., 2024), our Bayesian framework further shows that Mamba2 is an asymptotic special case of Palimpsa, where forgetting is so strong that computing the dynamic importance matrix becomes unnecessary. The fact that a continuum exists between Mamba2 and Palimpsa is an exciting direction for future work.

## 3 EXPERIMENTS

### 3.1 SYNTHETIC EXPERIMENTS: MAD

We evaluate Palimpsa on the Mechanistic Architecture Design (MAD) benchmark (Poli et al., 2024), a methodology for rapidly prototyping and testing sequence models. MAD utilizes a collection of synthetic tasks, such as recall, memorization, and compression, that serve as isolated "unit tests" to probe key architectural capabilities. MAD tasks are designed to identify which computational primitives excel at specific functions. Most importantly, the performance on the MAD benchmark were shown to correlate with compute-optimal performances on language tasks. For baselines, we used MAD benchmarks with default hyperparameters and grid search setting as published in (von Oswald et al., 2025). Palimpsa achieves a competitive average score compared to other models (Table 2). The model is among the best performers in almost every category, demonstrating particular strength in tasks related to state management. It obtains a perfect score on IC&Noisy Recall and delivers top-tier results on Memorize, Selective Copy, and Compress. The primary exception is Fuzzy

Table 2: Performance of Palimpsa on MAD benchmark (Poli et al., 2024). Others results are reported from (von Oswald et al., 2025).

| Model | IC& Noisy Recall | Fuzzy Recall | Memorize | Selective Copy | Compress | Average |
|-------|------------------|--------------|----------|----------------|----------|---------|
| Transformer | **100** | 48.6 | 84.7 | 96.0 | 49.5 | 75.8 |
| Mamba2 | **100** | 51.2 | 42.0 | 95.4 | 41.3 | 66.0 |
| GLA | **100** | 39.0 | 82.5 | 96.1 | 42.3 | 72.0 |
| xLSTM | **100** | 47.6 | 79.8 | 95.4 | 43.4 | 73.2 |
| DeltaNet | **100** | 55.5 | 40.8 | 98.8 | 43.3 | 67.7 |
| Gated DeltaNet | **100** | 32.7 | 81.7 | 95.7 | 45.0 | 71.0 |
| Hawk | 93.0 | 13.6 | **91.3** | 77.0 | 47.7 | 64.5 |
| MesaNet | **100** | **58.5** | 77.2 | 99.2 | 45.4 | **76.1** |
| Hawk-MesaNet | **100** | 30.2 | 85.6 | **99.6** | **52.3** | 73.5 |
| Palimpsa | **100** | 26.9 | 84.5 | 98.7 | 49.6 | 71.9 |

Recall, where its performance is lower. This indicates a potential trade-off between the model's capacity for precise state preservation and its ability to process more ambiguous information.

## 3.2 LANGUAGE MODELLING EXPERIMENTS

We pre-train Palimpsa and Gated Delta Networks on the fineweb-edu dataset for three different sizes and 15B and 30B tokens respectively. We evaluate Palimpsa on language modeling and common sense reasoning at academic scales. Following prior work (von Oswald et al., 2023a; Liu et al., 2024b; Yang et al., 2024), we test Wikitext perplexity and zero shot performance on a range of tasks such as LAMBADA (Paperno et al., 2016), PIQA (Bisk et al., 2020), HellaSwag (Zellers et al., 2019), WinoGrande (Sakaguchi et al., 2021), ARC (Clark et al., 2018), SIQA (Sap et al., 2019), and BoolQ (Clark et al., 2019). Results presented in Table 3, show that Palimpsa's performance relative to the Gated DeltaNet baseline (Yang et al., 2024) varies significantly with model scale. At the smallest scale of 170M parameters, Palimpsa is superior on all evaluated tasks. This advantage continues at the 340M scale, where Palimpsa maintains a better average score, but at the 760M scale, the Gated DeltaNet becomes the stronger model. This performance crossover suggests a complex interaction between our architecture and model scale. We propose two potential explanations for this scaling trend. The first is that the inductive biases from the metaplastic token mixer provide diminishing returns as model capacity increases, as the larger model may meta-learn other ways to manage state memories. The second is that the mechanism faces optimization challenges at a larger scale, preventing the model from fully leveraging its capabilities.

## 3.3 LIMITATIONS AND FUTURE WORK

A key limitation of our approach is its memory consumption, which impacts scalability. To compute the outputs for a given processing chunk, Palimpsa must explicitly materialize all intermediate states. This requirement leads to a significant memory footprint. This contrasts with architectures like Gated DeltaNet (Yang et al., 2024), which can compute all outputs for a given chunk using only its inputs and the final state of the previous chunk. Palimpsa cannot perform this direct computation because the element-wise product with the importance matrix $I_t$ is incompatible with the WY representation (Bischof and Van Loan, 1987) used by DeltaNet. To manage this memory demand, we are constrained to use smaller chunk dimensions, slowing training throughput for larger models (See Appendices). This finding suggests Palimpsa is best suited for smaller scales, a focus that is not a major drawback given the growing importance of small language models for agentic AI (Belcak et al., 2025). Based on our findings, several promising research directions emerge:

- The theoretical continuum identified between Palimpsa and Mamba2 could be explored further, for instance by fine-tuning Mamba2 models to incorporate metaplastic updates.

Table 3: Performance of Palimpsa and baselines on language modeling and common-sense reasoning tasks. The best results are highlighted. Results from Transformer++ at sizes 340M and 760M are reported from (Behrouz et al., 2024), while 170M is trained from scratch.

| Model | Wiki. ppl↓ | LMB. ppl↓ | LMB. acc↑ | PIQA acc↑ | Hella. acc_n↑ | Wino. acc↑ | ARC-e acc↑ | ARC-c acc_n↑ | SIQA acc↑ | BoolQ acc↑ | Avg. ↑ |
|---|---|---|---|---|---|---|---|---|---|---|---|
| 170M params / 15B tokens | | | | | | | | | | | |
| Transformer++ | 33.46 | 56.46 | **30.64** | 63.06 | 35.01 | 50.36 | 54.29 | 25.77 | 37.36 | 56.39 | 44.11 |
| Gated DeltaNet | 34.22 | 54.03 | 27.52 | 64.53 | 36.46 | 51.70 | 54.71 | 25.94 | 37.87 | 46.82 | 43.19 |
| Palimpsa | **32.16** | **52.32** | 27.91 | **66.59** | **37.44** | **52.25** | **56.44** | **26.71** | **38.84** | **59.91** | **45.76** |
| 340M params / 15B tokens | | | | | | | | | | | |
| Transformer++ | 31.52 | 41.08 | 30.76 | 62.98 | 34.76 | 50.53 | 45.21 | 24.05 | 36.81 | 58.24 | 42.92 |
| Gated DeltaNet | 27.72 | **37.65** | **30.97** | 66.49 | 39.89 | **51.46** | 58.75 | 27.47 | 38.69 | 54.28 | 46.00 |
| Palimpsa | **27.69** | 42.18 | 29.03 | **66.76** | **41.18** | 51.38 | **60.48** | **28.58** | **39.76** | **60.06** | **47.15** |
| 760M params / 30B tokens | | | | | | | | | | | |
| Transformer++ | 25.21 | 27.64 | 35.78 | 66.92 | 42.19 | 51.95 | 60.38 | 32.46 | 39.51 | 60.37 | 48.69 |
| Gated DeltaNet | **21.93** | **20.59** | **38.29** | 68.66 | **48.18** | **56.20** | 64.98 | 33.02 | **41.04** | **60.58** | **51.37** |
| Palimpsa | 22.96 | 23.43 | 36.64 | **68.72** | 46.84 | 52.64 | **65.45** | **33.28** | 39.56 | 59.02 | 50.27 |

- The output uncertainty from Palimpsa's Bayesian framework could be leveraged to improve overall model performance and reliability.

- To overcome current computational limitations, hybrid architectures could be developed, using Palimpsa for long-term memory and faster models for short-term processing.

## 4 CONCLUSION

We introduce Palimpsa, a novel attention mechanism derived from a Bayesian continual learning framework that incorporates principles of metaplasticity. By framing in-context learning as a continual learning problem, Palimpsa is designed to mitigate both catastrophic forgetting and remembering. Our experiments on commonsense reasoning and synthetic benchmarks demonstrate that Palimpsa achieves state-of-the-art performance, outperforming strong baselines like Gated DeltaNet, particularly at smaller model scales where its metaplastic inductive biases provide a distinct advantage.

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

## 5 APPENDICES

BAYESIAN FORGETTING

Following the work of (Bonnet et al., 2025b), we introduce a forgetting mechanism by considering a truncated posterior, which contains the last $N$ data points. The posterior over parameters $\boldsymbol{S}$ given data $\boldsymbol{d}_{t-N:t}$ is:

$$p(\boldsymbol{S}|\boldsymbol{d}_{t-N:t}) = \frac{p(\boldsymbol{d}_{t-N:t}|\boldsymbol{S})p(\boldsymbol{S})}{p(\boldsymbol{d}_{t-N:t})} = \frac{p(\boldsymbol{d}_t|\boldsymbol{S})p(\boldsymbol{S}|\boldsymbol{d}_{t-N:t-1})}{p(\boldsymbol{d}_t|\boldsymbol{d}_{t-N:t-1})}. \tag{4}$$

This can be expressed as a recursive update for a sliding window of size $N$. To move from the posterior over $\boldsymbol{d}_{t-N:t-1}$ to the one over $\boldsymbol{d}_{t-N+1:t}$, we incorporate the new data point $\boldsymbol{d}_t$ and remove the oldest one, $\boldsymbol{d}_{t-N}$:

$$p(\boldsymbol{S}|\boldsymbol{d}_{t-N+1:t}) \propto \underbrace{p(\boldsymbol{d}_t|\boldsymbol{S})p(\boldsymbol{S}|\boldsymbol{d}_{t-N:t-1})}_{\text{Learning}} \cdot \underbrace{\frac{1}{p(\boldsymbol{d}_{t-N}|\boldsymbol{S})}}_{\text{Forgetting}}. \tag{5}$$

In principle, one may not have access at time $t$ to the likelihood of the oldest data point, $p(\boldsymbol{d}_{t-N}|\boldsymbol{S})$. As a proxy, one can use the geometric mean of the likelihoods over the previous window, $\boldsymbol{d}_{t-N:t-1}$:

$$p(\boldsymbol{d}_{t-N}, \ldots, \boldsymbol{d}_{t-1}|\boldsymbol{S})^{\frac{1}{N}} \propto \left[ \frac{p(\boldsymbol{S}|\boldsymbol{d}_{t-N:t-1})}{p(\boldsymbol{S})} \right]^{\frac{1}{N}}. \tag{6}$$

This is made possible by approximating $p(\boldsymbol{S}|\boldsymbol{d}_{t-N:t-1})$, by the current variational truncated posterior $q_{\boldsymbol{\theta}_{t-1}}(\boldsymbol{S})$.

In simple terms, instead of completely discarding a single data point, this approach suggests we can discard a fraction of the joint likelihood of past data. This leads to a 'weighted' posterior where older data points have been discounted more heavily over time, keeping the total 'weight' of the posterior roughly constant (when $t \gg N$).

For Palimpsa, the forgetting needs to be input-dependent, similar to other gated recurrent models. Therefore, at each time step $t$, we discount the influence of all previous data by a factor related to a forgetting gate $\gamma_t \in [0, 1]$. The resulting weighted posterior at time $t$, denoted $p_w(\boldsymbol{S}|\boldsymbol{d}_{1:t})$, is defined recursively as:

$$p_w(\boldsymbol{S}|\boldsymbol{d}_{1:t}) \propto \underbrace{p(\boldsymbol{d}_t|\boldsymbol{S})p_w(\boldsymbol{S}|\boldsymbol{d}_{1:t-1})}_{\text{Learning}} \cdot \underbrace{\left( \frac{p_w(\boldsymbol{S}|\boldsymbol{d}_{1:t-1})}{p(\boldsymbol{S})} \right)^{-\frac{\gamma_t}{N}}}_{\text{Forgetting}}. \tag{7}$$

This is the target probability distribution that the variational distribution in Palimpsa approximates. Notably, this form of weighted posterior is the starting point for the general Bayesian framework for gated recurrent models.

VARIATIONAL FREE ENERGY OF THE WEIGHTED POSTERIOR

We minimize the Kullback–Leibler (KL) divergence between the variational distribution $q_{\boldsymbol{\theta}}(\boldsymbol{S}_i)$ and the weighted posterior $p_w(\boldsymbol{S_i}|\boldsymbol{d}_{1:t})$ by finding the optimal parameters $\boldsymbol{\theta}_{t,i}$:

$$\boldsymbol{\theta}_{t,i} = \arg\min_{\boldsymbol{\theta}} D_{\text{KL}}\left[ q_{\boldsymbol{\theta}}(\boldsymbol{S}_i) \, \| \, p_w(\boldsymbol{S_i}|\boldsymbol{d}_{1:t}) \right], \quad i = 1, \ldots, d_v.$$

Using the definition of the KL divergence, the recursive update from the previous section (Equation 7), and assuming that the previous posterior is well-approximated by our variational distribution, $p_w(\boldsymbol{S_i}|\boldsymbol{d}_{1:t-1}) \approx q_{\boldsymbol{\theta}_{t-1,i}}(\boldsymbol{S}_i)$, we can show that this is equivalent to minimizing the variational free energy $\mathcal{F}_{t,i}$:

$$\mathcal{F}_{t,i} = D_{\text{KL}}\left[ q_{\boldsymbol{\theta}_{t,i}}(\boldsymbol{S}_i) \, \| \, q_{\boldsymbol{\theta}_{t-1,i}}(\boldsymbol{S}_i) \right] - \mathbb{E}_{q_{\boldsymbol{\theta}_{t,i}}(\boldsymbol{S}_i)}\left[ \log p(\boldsymbol{d}_t|\boldsymbol{S}_i) - \frac{\gamma_t}{N} \log \frac{q_{\boldsymbol{\theta}_{t-1,i}}(\boldsymbol{S}_i)}{p(\boldsymbol{S}_i)} \right].$$

Now, using the standard formulas for Gaussian distributions (see Appendix) and considering only the terms that depend on the variational mean $\boldsymbol{\mu}_i$, we obtain the free energy expression discussed in the main text:

$$\mathcal{F}_{t,i}(\boldsymbol{\mu}_i) = \underbrace{\tfrac{1}{2}\|\boldsymbol{\mu}_i \boldsymbol{k}_t - v_{t,i}\|^2_{(\beta_{t,i})}}_{\text{plasticity}} + \underbrace{\frac{\gamma_t}{N}\frac{\boldsymbol{\mu}_i^2}{I_{prior}}}_{\text{forgetting}} + \underbrace{\left( 1 - \frac{\gamma_t}{N} \right)(\boldsymbol{\mu}_{t-1,i} - \boldsymbol{\mu}_i)^T \boldsymbol{\Sigma}_{t-1,i}^{-1}(\boldsymbol{\mu}_{t-1,i} - \boldsymbol{\mu}_i)}_{\text{stability}}.$$

However, for Palimpsa we are not only interested in the terms that depend on $\boldsymbol{\mu}_i$. First, let's derive the gradient with respect to all variational parameters.

**Calculating the Gradient of $\mathcal{F}_{t,i}$:** For readability, we first define the cost term $\mathcal{C}_{t,i}$ as the expected negative log-likelihood:

$$\mathcal{C}_{t,i} := -\mathbb{E}_{q_{\boldsymbol{\theta}_i}(\boldsymbol{S}_i)}\left[ \log p(v_{t,i}|\boldsymbol{k}_t, \beta_{t,i}, \boldsymbol{S}_i) \right] = \mathbb{E}_{\boldsymbol{\epsilon}_i}\left[ \mathcal{L}_i(\boldsymbol{S}_i) \right]. \tag{8}$$

To compute the partial derivatives of $\mathcal{C}_{t,i}$, we use the reparameterization trick: $\boldsymbol{S}_i = \boldsymbol{\mu}_i + \boldsymbol{A}_i \boldsymbol{\epsilon}_i$, where $\boldsymbol{\epsilon}_i \sim \mathcal{N}(0, \mathbb{I})$ and $\boldsymbol{A}_i \boldsymbol{A}_i^\top = \boldsymbol{\Sigma}_{t,i}$. This allows us to move the derivative operator inside the expectation:

$$\frac{\partial \mathcal{C}_i}{\partial \boldsymbol{\mu}_i} = \mathbb{E}_{\boldsymbol{\epsilon}_i} \left[ \frac{\partial \mathcal{L}_i(\boldsymbol{S}_i)}{\partial \boldsymbol{S}_i} \right] \qquad \frac{\partial \mathcal{C}_i}{\partial \boldsymbol{A}_i} = \mathbb{E}_{\boldsymbol{\epsilon}_i} \left[ \frac{\partial \mathcal{L}_i(\boldsymbol{S}_i)}{\partial \boldsymbol{S}_i} \boldsymbol{\epsilon}_i^\top \right]. \tag{9}$$

Generally, these gradients would be estimated using Monte Carlo sampling. However, in Palimpsa we are in the case of a linear regression problem so the gradients can be calculated analytically. Substituting the derivative of the linear regression log-likelihood gives:

$$\frac{\partial \mathcal{C}_i}{\partial \boldsymbol{\mu}_i} = \mathbb{E}_{\boldsymbol{\epsilon}_i} \left[ \beta_{t,i} \left( (\boldsymbol{\mu}_i + \boldsymbol{A}_i \boldsymbol{\epsilon}_i)^\top \boldsymbol{k}_t - v_{t,i} \right) \boldsymbol{k}_t \right] \qquad \frac{\partial \mathcal{C}_i}{\partial \boldsymbol{A}_i} = \mathbb{E}_{\boldsymbol{\epsilon}_i} \left[ \beta_{t,i} \left( (\boldsymbol{\mu}_i + \boldsymbol{A}_i \boldsymbol{\epsilon}_i)^\top \boldsymbol{k}_t - v_{t,i} \right) \boldsymbol{k}_t \boldsymbol{\epsilon}_i^\top \right]. \tag{10}$$

Knowing that $\mathbb{E}_{\boldsymbol{\epsilon}_i}[\boldsymbol{\epsilon}_i] = \vec{0}$ and $\mathbb{E}_{\boldsymbol{\epsilon}_i}[\boldsymbol{\epsilon}_i \boldsymbol{\epsilon}_i^\top] = \mathbb{I}$ (the identity matrix), we can resolve the expectations to get the final analytical gradients of the cost term:

$$\frac{\partial \mathcal{C}_i}{\partial \boldsymbol{\mu}_i} = \beta_{t,i} \left( \boldsymbol{k}_t \boldsymbol{k}_t^\top \boldsymbol{\mu}_i - v_{t,i} \boldsymbol{k}_t \right) \qquad \frac{\partial \mathcal{C}_i}{\partial \boldsymbol{A}_i} = \beta_{t,i} \boldsymbol{k}_t \boldsymbol{k}_t^\top \boldsymbol{A}_i. \tag{11}$$

Finally, we add the gradients from the other terms in the free energy objective (the KL-divergence and forgetting terms) to obtain the full gradients of $\mathcal{F}_{t,i}$:

$$\frac{\partial \mathcal{F}_{t,i}}{\partial \boldsymbol{\mu}_i} = \left( 1 - \frac{\gamma_t}{N} \right) \boldsymbol{\Sigma}_{t-1,i}^{-1} (\boldsymbol{\mu}_i - \boldsymbol{\mu}_{t-1,i}) + \frac{\gamma_t}{N} \boldsymbol{\mu}_i + \beta_{t,i} \left( \boldsymbol{k}_t \boldsymbol{k}_t^\top \boldsymbol{\mu}_i - v_{t,i} \boldsymbol{k}_t \right), \tag{12}$$

$$\frac{\partial \mathcal{F}_{t,i}}{\partial \boldsymbol{A}_i} = \left[ \left( 1 - \frac{\gamma_t}{N} \right) \boldsymbol{\Sigma}_{t-1,i}^{-1} + \frac{\gamma_t}{N} \boldsymbol{I}_{prior} \right] \boldsymbol{A}_i - (\boldsymbol{A}_i^{-1})^\top + \beta_{t,i} \boldsymbol{k}_t \boldsymbol{k}_t^\top \boldsymbol{A}_i. \tag{13}$$

**Closed-Form Solution of $\mathcal{F}_{t,i}$:** To find the optimal variational parameters, we set the gradients of the free energy $\mathcal{F}_{t,i}$ to zero and solve.

First, to find the optimal covariance matrix $\boldsymbol{\Sigma}_{t,i}$, we solve for $\boldsymbol{A}_{t,i}$ in the equation:

$$\frac{\partial \mathcal{F}_{t,i}}{\partial \boldsymbol{A}_{t,i}} = \vec{0}. \tag{14}$$

Setting the gradient expression from the previous section to zero gives:

$$\left[ \left( 1 - \frac{\gamma_t}{N} \right) \boldsymbol{\Sigma}_{t-1,i}^{-1} + \frac{\gamma_t}{N} \boldsymbol{I}_{prior} + \beta_{t,i} \boldsymbol{k}_t \boldsymbol{k}_t^\top \right] \boldsymbol{A}_{t,i} - (\boldsymbol{A}_{t,i}^{-1})^\top = \boldsymbol{0}. \tag{15}$$

To solve for the full covariance matrix $\boldsymbol{\Sigma}_{t,i} = \boldsymbol{A}_{t,i} \boldsymbol{A}_{t,i}^\top$, we can multiply on the right by $\boldsymbol{A}_{t,i}^\top$:

$$\left[ \left( 1 - \frac{\gamma_t}{N} \right) \boldsymbol{\Sigma}_{t-1,i}^{-1} + \frac{\gamma_t}{N} \boldsymbol{I}_{prior} + \beta_{t,i} \boldsymbol{k}_t \boldsymbol{k}_t^\top \right] \boldsymbol{\Sigma}_{t,i} - \boldsymbol{I} = \boldsymbol{0}. \tag{16}$$

Rearranging the terms, we find that the new precision matrix (the inverse covariance) is a sum of the discounted old precision, the prior precision, and the new data term:

$$\boldsymbol{\Sigma}_{t,i}^{-1} = \left( 1 - \frac{\gamma_t}{N} \right) \boldsymbol{\Sigma}_{t-1,i}^{-1} + \frac{\gamma_t}{N} \boldsymbol{I}_{prior} + \beta_{t,i} \boldsymbol{k}_t \boldsymbol{k}_t^\top. \tag{17}$$

The closed-form update for the covariance is the inverse of this expression:

$$\boxed{\boldsymbol{\Sigma}_{t,i} = \left[ \left( 1 - \frac{\gamma_t}{N} \right) \boldsymbol{\Sigma}_{t-1,i}^{-1} + \frac{\gamma_t}{N} \boldsymbol{I}_{prior} + \beta_{t,i} \boldsymbol{k}_t \boldsymbol{k}_t^\top \right]^{-1}} \tag{18}$$

The major challenge in this analytical approach is to compute the $d_k \times d_k$ matrix inverse at each time step. In the case without forgetting ($N \to \infty$), (von Oswald et al., 2019) use the Sherman-Morrison formula to compute it recursively, but this approach did not scale well. Another approach is to use a parallel conjugate gradient method to solve the associated linear systems, as done in

MesaNet (von Oswald et al., 2023a). To connect this with the notation in MesaNet (von Oswald et al., 2023a), our precision matrix $\boldsymbol{\Sigma}_t^{-1}$ corresponds to their $\boldsymbol{H}_t + \boldsymbol{\Lambda}$, and our precision-weighted mean $\boldsymbol{\Sigma}_t^{-1}\boldsymbol{\mu}_t$ corresponds to their $\boldsymbol{G}_t$. In their work, the precision matrix is identical for all rows $i$ (i.e., $\boldsymbol{\Sigma}_{t,i}^{-1} = \boldsymbol{\Sigma}_t^{-1}$) because their $\beta_t$ is a scalar.

Similarly, we find the optimal mean $\boldsymbol{\mu}_{t,i}$ by setting its corresponding gradient to zero:

$$\frac{\partial \mathcal{F}_{t,i}}{\partial \boldsymbol{\mu}_{t,i}} = \vec{0}. \tag{19}$$

$$\left(1 - \frac{\gamma_t}{N}\right) \boldsymbol{\Sigma}_{t-1,i}^{-1}(\boldsymbol{\mu}_{t,i} - \boldsymbol{\mu}_{t-1,i}) + \frac{\gamma_t}{N} \boldsymbol{I}_{prior}\boldsymbol{\mu}_{t,i} + \beta_{t,i}\left(\boldsymbol{k}_t\boldsymbol{k}_t^\top \boldsymbol{\mu}_{t,i} - v_{t,i}\boldsymbol{k}_t\right) = \vec{0}. \tag{20}$$

Grouping the terms with $\boldsymbol{\mu}_{t,i}$ yields:

$$\left[\left(1 - \frac{\gamma_t}{N}\right)\boldsymbol{\Sigma}_{t-1,i}^{-1} + \frac{\gamma_t}{N}\boldsymbol{I}_{prior} + \beta_{t,i}\boldsymbol{k}_t\boldsymbol{k}_t^\top\right]\boldsymbol{\mu}_{t,i} = \left(1 - \frac{\gamma_t}{N}\right)\boldsymbol{\Sigma}_{t-1,i}^{-1}\boldsymbol{\mu}_{t-1,i} + \beta_{t,i}v_{t,i}\boldsymbol{k}_t. \tag{21}$$

Recognizing the term in brackets as the new precision $\boldsymbol{\Sigma}_{t,i}^{-1}$, we arrive at the solution for the mean:

$$\boxed{\boldsymbol{\mu}_{t,i} = \boldsymbol{\Sigma}_{t,i}\left[\left(1 - \frac{\gamma_t}{N}\right)\boldsymbol{\Sigma}_{t-1,i}^{-1}\boldsymbol{\mu}_{t-1,i} + \beta_{t,i}v_{t,i}\boldsymbol{k}_t\right]} \tag{22}$$

DERIVATION OF PALIMPSA

In Palimpsa, we aim to solve the update equations with a vector input gating $\boldsymbol{\beta}_t \in \mathbb{R}^{d_v}$, where each row of parameters has its own rate. This change prevents a simple closed-form solution for the matrix inverse. To make the problem tractable and computationally efficient, we introduce a diagonal approximation for the precision matrix. By assuming the precision matrices are diagonal, $\boldsymbol{\Sigma}_{t,i}^{-1} = \text{diag}(\boldsymbol{I}_{t,i})$, we can apply this approximation to the closed-form solutions from the previous section. This yields element-wise update rules for the diagonal precision vector $\boldsymbol{I}_{t,i}$ and the mean vector $\boldsymbol{\mu}_{t,i}$ for each row $i$:

$$\boldsymbol{I}_{t,i} = \left(1 - \frac{\gamma_t}{N}\right)\boldsymbol{I}_{t-1,i} + \frac{\gamma_t}{N}\boldsymbol{I}_{prior} + \beta_{t,i}\boldsymbol{k}_t^2 \tag{23}$$

$$\boldsymbol{\mu}_{t,i} = \left(1 - \frac{\gamma_t}{N}\right)\frac{\boldsymbol{I}_{t-1,i}}{\boldsymbol{I}_{t,i}} \odot \boldsymbol{\mu}_{t-1,i} + \frac{1}{\boldsymbol{I}_{t,i}} \odot (\beta_{t,i}v_{t,i}\boldsymbol{k}_t) \tag{24}$$

where $\boldsymbol{k}_t^2$ denotes the element-wise square of $\boldsymbol{k}_t$, and all divisions are element-wise. Note that this is a stronger approximation than a standard mean-field (diagonal covariance) assumption because we derived the fully-coupled solution first and only then discarded the off-diagonal terms. This approach is similar to the "diagonal approximation" used in Longhorn (Liu et al., 2024b) and is equivalent to treating each parameter (or "synapse") as an independent Gaussian distribution.

From here, by defining the forgetting factor $\alpha_t = (1 - \frac{\gamma_t}{N})$ and stacking the row vectors into matrices, we can write the final updates for Palimpsa in matrix form:

$$\boxed{\boldsymbol{I}_t = \alpha_t\boldsymbol{I}_{t-1} + (1 - \alpha_t)\boldsymbol{I}_{prior} + \boldsymbol{\beta}_t \otimes \boldsymbol{k}_t^2} \tag{25}$$

$$\boxed{\boldsymbol{\mu}_t = \alpha_t\frac{\boldsymbol{I}_{t-1}}{\boldsymbol{I}_t} \odot \boldsymbol{\mu}_{t-1} + \frac{1}{\boldsymbol{I}_t} \odot \left[(\boldsymbol{\beta}_t \odot \boldsymbol{v}_t) \otimes \boldsymbol{k}_t\right]} \tag{26}$$

DERIVATION OF DELTANET AND GATED DELTANET

Let's take the case of Gated DeltaNet, as one just has to suppress the forgetting for the standard DeltaNet. Starting from the free energy equation and taking $\beta_t \in \mathbb{R}$:

$$\mathcal{F}_{t,i}(\boldsymbol{\mu}_i) = \underbrace{\tfrac{1}{2}\|\boldsymbol{\mu}_i^\top\boldsymbol{k}_t - v_{t,i}\|_{\beta_t}^2}_{\text{plasticity}} + \underbrace{\tfrac{1}{2}\frac{\gamma_t}{N}\frac{\|\boldsymbol{\mu}_i\|^2}{I_{prior}}}_{\text{forgetting}} + \underbrace{\tfrac{1}{2}\left(1 - \frac{\gamma_t}{N}\right)(\boldsymbol{\mu}_{t-1,i} - \boldsymbol{\mu}_i)^\top\boldsymbol{\Sigma}_{t-1,i}^{-1}(\boldsymbol{\mu}_{t-1,i} - \boldsymbol{\mu}_i)}_{\text{stability}},$$

with the simplifications $\boldsymbol{\Sigma}_{t-1,i}^{-1} = \mathbb{I}_{d_k}$ and $I_{prior} = 1$. If we consider the gradient of the plasticity term to be linear around $\boldsymbol{\mu}_{t-1,i}$ (a first-order approximation), we assume that:

$$\beta_t \left( \boldsymbol{k}_t \boldsymbol{k}_t^\top \boldsymbol{\mu}_i - v_{t,i} \boldsymbol{k}_t \right) \approx \beta_t \left( \boldsymbol{k}_t \boldsymbol{k}_t^\top \boldsymbol{\mu}_{t-1,i} - v_{t,i} \boldsymbol{k}_t \right).$$

Then, setting the full gradient to zero, $\frac{\partial \mathcal{F}_{t,i}}{\partial \boldsymbol{\mu}_i} = \vec{0}$, is given by:

$$\left( 1 - \frac{\gamma_t}{N} \right) \mathbb{I}_{d_k} (\boldsymbol{\mu}_i - \boldsymbol{\mu}_{t-1,i}) + \frac{\gamma_t}{N} \boldsymbol{\mu}_i + \beta_t \left( \boldsymbol{k}_t \boldsymbol{k}_t^\top \boldsymbol{\mu}_{t-1,i} - v_{t,i} \boldsymbol{k}_t \right) = \vec{0}. \tag{27}$$

Solving for $\boldsymbol{\mu}_i$ yields:

$$\boldsymbol{\mu}_i = \left( 1 - \frac{\gamma_t}{N} \right) \boldsymbol{\mu}_{t-1,i} + \beta_t \left( \boldsymbol{k}_t \boldsymbol{k}_t^\top \boldsymbol{\mu}_{t-1,i} - v_{t,i} \boldsymbol{k}_t \right). \tag{28}$$

From there, by defining $\alpha_t = (1 - \frac{\gamma_t}{N})$, we can write the final solution in matrix form as:

$$\boldsymbol{\mu}_t = \boldsymbol{\mu}_{t-1} \left( \alpha_t \mathbb{I}_{d_k} + \beta_t \boldsymbol{k}_t \boldsymbol{k}_t^\top \right) - \beta_t \boldsymbol{v}_t \boldsymbol{k}_t^\top. \tag{29}$$

This is equivalent to Gated DeltaNet if its input gate is $\beta_t^{gdn} = \frac{\beta_t}{\alpha_t}$, and the same as the standard DeltaNet when $N \to \infty$, i.e., $\alpha_t \to 1$.

GAUSSIAN CHEAT SHEET

**Entropy**   The entropy of a multivariate Gaussian distribution $q_{\boldsymbol{\theta}_1}(\boldsymbol{S}_i) = \mathcal{N}(\boldsymbol{\mu}_1, \boldsymbol{\Sigma}_1)$ of dimension $d_k$ is given by:

$$H(q_{\boldsymbol{\theta}_1}) = -\mathbb{E}_{q_{\boldsymbol{\theta}_1}(\boldsymbol{S}_i)} \left[ \log q_{\boldsymbol{\theta}_1}(\boldsymbol{S}_i) \right]$$

$$H(q_{\boldsymbol{\theta}_1}) = \frac{d_k}{2} \log(2\pi e) + \frac{1}{2} \log \det(\boldsymbol{\Sigma}_1)$$

**KL Divergence**   The KL divergence between two multivariate Gaussian distributions $q_{\boldsymbol{\theta}_1}(\boldsymbol{S}_i) = \mathcal{N}(\boldsymbol{\mu}_1, \boldsymbol{\Sigma}_1)$ and $q_{\boldsymbol{\theta}_2}(\boldsymbol{S}_i) = \mathcal{N}(\boldsymbol{\mu}_2, \boldsymbol{\Sigma}_2)$ is given by:

$$D_{\text{KL}} \left[ q_{\boldsymbol{\theta}_1}(\boldsymbol{S}_i) \,||\, q_{\boldsymbol{\theta}_2}(\boldsymbol{S}_i) \right] = \mathbb{E}_{q_{\boldsymbol{\theta}_1}(\boldsymbol{S}_i)} \left[ \log \frac{q_{\boldsymbol{\theta}_1}(\boldsymbol{S}_i)}{q_{\boldsymbol{\theta}_2}(\boldsymbol{S}_i)} \right]$$

$$D_{\text{KL}} \left[ q_{\boldsymbol{\theta}_1}(\boldsymbol{S}_i) \,||\, q_{\boldsymbol{\theta}_2}(\boldsymbol{S}_i) \right] = \frac{1}{2} \left[ \text{tr}(\boldsymbol{\Sigma}_2^{-1} \boldsymbol{\Sigma}_1) + (\boldsymbol{\mu}_2 - \boldsymbol{\mu}_1)^T \boldsymbol{\Sigma}_2^{-1} (\boldsymbol{\mu}_2 - \boldsymbol{\mu}_1) - d_k + \log \frac{\det \boldsymbol{\Sigma}_2}{\det \boldsymbol{\Sigma}_1} \right]$$

**Cross-Entropy**   The cross-entropy between two multivariate Gaussian distributions $q_{\boldsymbol{\theta}_1}(\boldsymbol{S}_i) = \mathcal{N}(\boldsymbol{\mu}_1, \boldsymbol{\Sigma}_1)$ and $q_{\boldsymbol{\theta}_2}(\boldsymbol{S}_i) = \mathcal{N}(\boldsymbol{\mu}_2, \boldsymbol{\Sigma}_2)$ is given by:

$$H(q_{\boldsymbol{\theta}_1}, q_{\boldsymbol{\theta}_2}) = -\mathbb{E}_{q_{\boldsymbol{\theta}_1}(\boldsymbol{S}_i)} \left[ \log q_{\boldsymbol{\theta}_2}(\boldsymbol{S}_i) \right]$$

It can be found using the relation:

$$H(q_{\boldsymbol{\theta}_1}, q_{\boldsymbol{\theta}_2}) = H(q_{\boldsymbol{\theta}_1}) + D_{\text{KL}} \left[ q_{\boldsymbol{\theta}_1}(\boldsymbol{S}_i) \,||\, q_{\boldsymbol{\theta}_2}(\boldsymbol{S}_i) \right]$$

The final expression is:

$$H(q_{\boldsymbol{\theta}_1}, q_{\boldsymbol{\theta}_2}) = \frac{1}{2} \left[ \text{tr}(\boldsymbol{\Sigma}_2^{-1} \boldsymbol{\Sigma}_1) + (\boldsymbol{\mu}_2 - \boldsymbol{\mu}_1)^T \boldsymbol{\Sigma}_2^{-1} (\boldsymbol{\mu}_2 - \boldsymbol{\mu}_1) + d_k \log(2\pi) + \log \det \boldsymbol{\Sigma}_2 \right]$$

MAD EXPERIMENTS

We follow the benchmarking procedure detailed in (Poli et al., 2024), precisely: For each task in the suite, we evaluate the architectures on subtasks of varying difficulty (i.e. varying sequence length, number of training examples, vocabulary sizes and further, task-specific parameters) and compute the mean accuracy. We further sweep over varying learning rates and weight decay values for each model and report the maximum average task accuracy. For each architecture, we fix a set of hyper-parameters following the work of (von Oswald et al., 2025) that can be found in Table 4.

| Hyper Parameter | Value / Search Space |
|---|---|
| Embedding dimension | 128 |
| Number of layers | 2 |
| Number of heads | 8 |
| Value expansion factor ($expandv$) | 1 |
| Key expansion factor ($expandk$) | 1 |
| Initial state memory ($N_{init}$) | 32 |
| $\log N$ initialization | $\mathcal{U}(-\log(4), \log(4))$ across heads |
| Epochs | 200 |
| Batch size | 32 |
| Optimizer | AdamW |
|    Learning rate | [3e-3, 1e-3, 5e-4, 1e-4] |
|    Weight decay | [0.01, 0.1] |
|    $\beta$s | (0.9, 0.98) |
| Scheduler | Cosine Scheduler with Warmup |
|    Minimum learning rate | 1e-5 |
|    Warm-up start learning rate | 1e-7 |
|    Warm-up steps | 750 |

Table 4: MAD benchmark suite hyper-parameters, taken from (Poli et al., 2024). Model-specific parameters for Palimpsa are included.

Table 5: Architectural and training details for language modeling experiments. Key and value expansion factors are denoted by $E_k$ and $E_v$ respectively.

| Model Size | Model Name | Layers | Dim | Heads | $E_k$ | $E_v$ | $N_{init}$ | Peak LR | Tokens |
|---|---|---|---|---|---|---|---|---|---|
| 170M | Transformer++ | 20 | | | – | – | – | | |
| | Gated DeltaNet | 19 | 768 | 16 | 1.0 | 1.0 | – | 3e-3 | 15B |
| | Palimpsa | 21 | | | 0.5 | 1.0 | 16 | | |
| 340M | Gated DeltaNet | 22 | 1024 | 16 | 1.0 | 1.0 | – | 1.5e-3 | 15B |
| | Palimpsa | 24 | | | 0.5 | 1.0 | 16 | | |
| 760M | Gated DeltaNet | 23 | 1536 | 16 | 1.0 | 1.0 | – | 1.25e-3 | 30B |
| | Palimpsa | 25 | | | 0.5 | 1.0 | 16 | | |

LANGUAGE MODELLING EXPERIMENTS

The architectural and training details for Palimpsa, Gated DeltaNet, and Transformer++ are found in Table 5. For Palimpsa, the $\log N$ parameter is initialized for each of the 16 heads by sampling from a uniform distribution $\mathcal{U}(-\log(4), \log(4))$. Note that for all the model we tied words embedding. For Palimpsa, we choose the state size to be half that of Gated Deltanet ($E_k$ is divided by two compared to Gated DeltaNet) so that the state sizes remain comparable.

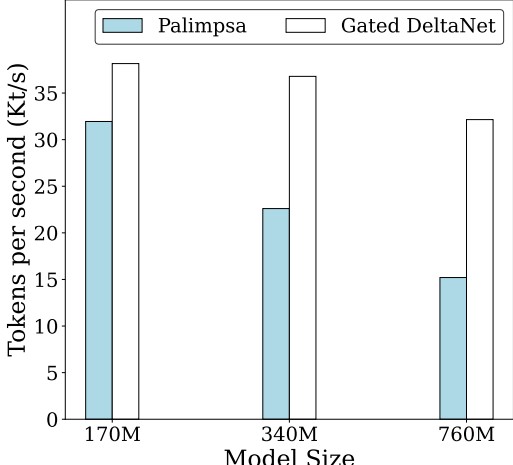

Figure 2: Training throughput of 1.3B models on a single H100, the relative performance of Palimpsa compared to Gated DeltaNet decrease with model size

PALIMPSA IMPLEMENTATION AND PARALLELIZATION

Our parallel training algorithm is implemented as a fused Triton kernel using a chunk-wise scan. The core logic is as follows:

- **Chunk-wise Processing:** The input sequence is split into chunks of size $B_T$.
- **Intra-Chunk Parallel Scan:** Within each chunk, the recursive state updates for $\boldsymbol{\mu}_t$ and $\boldsymbol{I}_t$ are computed in parallel using a cumulative sum operation.
- **Inter-Chunk Sequential Update:** The final state of one chunk is carried over as the initial state for the next.

This approach allows for efficient GPU utilization during training. The kernel also performs state checkpointing to ensure numerical stability during the backward pass.

To implement the parallel scan, the recurrences are reformulated. Let $t \in [1, B_T]$ be the index within a chunk, and let $\boldsymbol{M}_0$ and $\boldsymbol{I}_0$ be the final states from the previous chunk. The states at every position $t$ in the current chunk are computed as:

$$\boldsymbol{A}_t = \operatorname*{cumprod}_{j=1...t} (\alpha_j) \tag{30}$$

$$\boldsymbol{A}_t^r = \operatorname*{reverse\text{-}cumprod}_{j=1...t} (\alpha_j) \tag{31}$$

$$\boldsymbol{M}_t = \boldsymbol{A}_t^r \boldsymbol{M}_0 \boldsymbol{I}_0 + \operatorname*{cumsum}_{j=1...t} ((\boldsymbol{\beta}_j \odot \boldsymbol{v}_j) \otimes \boldsymbol{k}_j \boldsymbol{A}_{t,j}) / \boldsymbol{A}_t \tag{32}$$

$$\boldsymbol{I}_t = \boldsymbol{A}_t^r \boldsymbol{I}_0 + (1 - \boldsymbol{A}_t^r)\boldsymbol{I}_{\text{prior}} + \operatorname*{cumsum}_{j=1...t} \left(\boldsymbol{\beta}_j \otimes \boldsymbol{k}_j^2 \boldsymbol{A}_{t,j}\right) / \boldsymbol{A}_t \tag{33}$$

The final state is then recovered by $\boldsymbol{\mu}_t = \boldsymbol{M}_t / \boldsymbol{I}_t$. The cumulative product in Eq. 31 is computed efficiently in log-space, also as a cumulative sum.

The main limitations Palimpsa must explicitly materialize all intermediate states. This requirement leads to a significant memory footprint. To manage this memory demand, we are constrained to use smaller chunk dimensions, slowing training throughput for larger models as shown in Fig. 2.