# OpenReview forum: "Learning to Remember, Learn, and Forget in Attention-Based Models"
_ICLR.cc/2026/Conference — ICLR 2026 Conference Withdrawn Submission_

### Official Review · Reviewer_HigJ · 2025-10-26

**Soundness:** 2
**Presentation:** 2
**Contribution:** 2
**Rating:** 4
**Confidence:** 3

**Summary:**

The paper introduces Palimpsa, a new attention mechanism that treats memory updates as Bayesian inference. Inspired by metaplasticity in neuroscience, Palimpsa maintains both a memory mean and an uncertainty parameter for each attention state. The paper hypothesizes that this allows the model to adaptively balance learning new information, retaining important past content, and forgetting outdated inputs. The authors derive closed-form update rules from a variational free-energy objective, unifying prior models like Mamba2 and MesaNet as special cases in this framework. Experiments on synthetic memory benchmarks and language tasks show that Palimpsa achieves comparable performance to previous baselines.

**Strengths:**

The main strength is that the paper provides a compelling Bayesian reinterpretation of attention and state-space models, showing that many recent architectures (e.g., Gated DeltaNet, Mamba2, MesaNet) can be viewed as special cases of a single variational inference framework. This theoretical unification helps clarify connections between disparate model classes and gives principled meaning to heuristic gating mechanisms used in prior work.

**Weaknesses:**

1. Some parts of the paper are not well-presented. In Section 2.2, β is not defined until several paragraphs later and initially reads as if it comes directly from the input. The notation of placing diag(β) in a subscript is also unfamiliar to me and likely to other readers. It was not immediately clear why or how we aim to maximize the objective on line 245 — if I understand correctly, the goal is to update S to accommodate the new key–value pair (k, v). On line 265, what is x and how is it linked to k and v? The variational approximation introduced on line 273 also lacks context; at minimum, the authors should mention that p is intractable. I suggest clarifying these points, as they appear to form the core of the proposed method.

2. The model’s performance does not fully convince me that Palimpsa is a stronger method. On the MAD benchmark, Palimpsa performs worse than Gated Linear Attention and MesaNet. For language modeling, the set of baselines appears somewhat limited. Moreover, the performance gain reverses for larger networks, as the authors themselves acknowledge.

3. The core idea of Palimpsa is to adaptively update the attention state based on the current input. However, it remains unclear how memory and forgetting actually operate in practice. Including visualizations, even on toy tasks, could help illustrate these dynamics. The paper also lacks ablation studies, making it difficult to assess under what conditions the proposed update rule is effective.

4. The link to metaplasticity in neuroscience is mainly conceptual. Context-modulated plasticity is not new and has previously been explored in recurrent neural networks [1, 2].

[1] Backpropamine: Training self-modifying neural networks with differentiable neuromodulated plasticity
[2] Hebbian and Gradient-based Plasticity Enables Robust Memory and Rapid Learning in RNNs

**Questions:**

1. How important is the adaptive forgetting and momorization mechanism (metaplsticity) in Palimpsa? If we just use a constant $\gamma$, or use a constant learnable vector $\beta$ that does not change for each step $t$, how does that affect empirical performance?

2. While the comparisons in Table 1 is very interesting, it is not clear to me what are the weaknesses of previous models that Palimpsa attempt to address.

3. The authors mention that Palimpsa uses the simplification that uses a diagonized stability matrix, could the authors discuss the implication for such simplification?

4. How does Palimpsa decide when to forget versus consolidate? Is the forgetting gate driven purely by input statistics, or does it evolve based on longer-term task structure?

5. How would you interpret the inconsistent results on different MAD subtasks? For example, why would Palimpsa less well on Fuzzy recall?

---

> ### Author Response · Authors · 2025-12-03
> **Rebuttal**
>
> Thank you for your encouraging review. We are pleased you found the Bayesian reinterpretation of attention and state-space models compelling.
>
> 1. **Notation and Tractability:** We appreciate the detailed feedback on notation. To clarify: $\mathbf{x}$ is the layer input, while $\mathbf{k}, \mathbf{v}, \mathbf{q}, \boldsymbol{\beta}$ are projections of $\mathbf{x}$. Crucially, regarding the variational approximation: in this linear regression formulation with Gaussian priors, the true posterior $p$ is analytically tractable and Gaussian. Therefore, Variational Inference yields the *exact* solution here, not an approximation. We utilized the VI framework specifically to derive the free-energy objective, which serves as the unifying link between different models in the literature. We will ensure this is clarified in future revisions.
>
> 2. **Performance:** We maintain that Palimpsa is a stronger model specifically for smaller scales where efficient state management is critical. We acknowledge that the current results on the MAD benchmark and larger networks indicate limitations that require further investigation. However, regarding the MAD benchmark, we observed that performance on Fuzzy Recall is highly sensitive to parameters unrelated to the token mixer, which diminishes the reliability of these results as a comparative metric.
>
> 3. **Metaplasticity vs. Context-Modulated Plasticity:** We distinguish between context-modulated plasticity (achieved via input gating $\boldsymbol{\beta}$) and metaplasticity, where the plasticity (learning rate) is itself a dynamic state ($\mathbf{I}_t$). In Palimpsa, the precision matrix acts as a dynamic, per-parameter learning rate. While the concept of metaplasticity exists in the literature, Palimpsa applies it specifically to the token mixer of attention-based models, deriving a Hebbian update rule where weight changes are scaled by their uncertainty. Backpropamine can indeed be related to attention updates in in-context learning and the "meta-learning" view of transformers, but not to metaplasticity. Metaplasticity goes beyond this concept by effectively modifying the "learning rate" of the attention weight updates depending on task importance.

---

### Official Review · Reviewer_jHq3 · 2025-10-29

**Soundness:** 2
**Presentation:** 3
**Contribution:** 2
**Rating:** 2
**Confidence:** 3

**Summary:**

The authors frame in-context learning as an online continual learning problem – with the concomitant tension between retaining past and absorbing new information – and propose a Bayesian framework with which to understand the trade-off. They apply the framework to several linear-gated transformers and use it to derive their own metaplasticity-inspired layer. Results on language modeling, commonsense reasoning, and the MAD benchmark are presented. The authors state that they have implemented custom kernels to speed up the computation of their layer.

**Strengths:**

The attempt to formally unify linear-gated transformers is admirable (see the Appendix) and a common area of recent work. This reviewer finds the framing of in-context learning as a continual learning problem compelling. The quality of the writing is above average and the arguments are typically clear.

**Weaknesses:**

This reviewer believes the evaluation and results of the authors’ layer, Palimpsa, could be much stronger. The language modeling and commonsense reasoning evaluation only compares to Gated DeltaNet. We have observed that results on those tasks tend to be much better with Gated DeltaNet-H2. While the authors give DeltaNet and Gated DeltaNet special treatment in the Appendix, that should not preclude the use of a different model – one closer to state-of-the-art – in their evaluation. The results on the synthetic MAD benchmark place Palimpsa in the middle of the pack (itself not necessarily a problem) and, sadly, the results on Fuzzy Recall and Memorize – tasks on which one would expect Palimpsa to excel due to the metaplasticity-inspired design of the layer – are underwhelming. I would find this paper more compelling if the evaluation contained, say, a substantial amount of error analysis or other phenomenology that would demonstrate the failure modes of Palimpsa. Such analysis can bring substantial value to a report when the results are not state of the art and, especially, when the results are not state of the art on tasks on which the model should be quite superior.

Footnote 2 states that the code (or the link thereto) has been omitted to ensure double-blind review. This reviewer reminds the authors that sites such as https://anonymous.4open.science/ support anonymous repositories.

There are a number of strange paragraph breaks in the paper. I would encourage the authors to track down the cause.

**Questions:**

This reviewer invites the authors to respond to any of the points I have raised in the weaknesses section above.

---

> ### Author Response · Authors · 2025-12-03
> **Rebuttal**
>
> Thank you for your encouraging review. We are pleased you found compelling the continual learning interpretation of in-context learning.
>
> 1. **Baselines:** Gated DeltaNet-H2 is a hybrid model, whereas we focus solely on linear models. Introducing hybrid models into the comparison would require doing the same for Palimpsa. While an interesting avenue, we decided that this was out of the scope of this article.
>
> 2. **MAD Results:** We agree that the results on Fuzzy Recall are disappointing; one would indeed expect Palimpsa to perform better on this benchmark. Despite our best efforts, we have not yet pinpointed the reasons for the poor results on this task. We noticed this benchmark is extremely sensitive to weight initialization and architecture choices. We note, however, that the Memorize task does not involve in-context learning; rather, it targets the memorization of factual knowledge.
>
> 2. **Reproducibility:** The anonymous repository was indeed an option. However, the supplementary material already provided the necessary kernel files for reproducibility.

---

### Official Review · Reviewer_dUCE · 2025-10-31

**Soundness:** 2
**Presentation:** 2
**Contribution:** 2
**Rating:** 4
**Confidence:** 4

**Summary:**

The paper proposes Palimpsa, a dual‑state, fixed‑memory token mixer that casts self‑attention as online Bayesian inference with metaplasticity. Concretely, it maintains a per‑state Gaussian posterior over the attention memory with a diagonal precision vector  I_t, yielding closed‑form, per‑token updates that gate input, stability, and forgetting; forgetting is tied to input, so no forgetting results in no input. The authors argue this framework unifies several gated models and show Mamba2 as an asymptotic special case under strong forgetting.

**Strengths:**

1. Principled derivation. The free‑energy view leads to exact, closed‑form update rules (Eq. 3) for both the mean and the per‑synapse importance, providing a clean, test‑time learning interpretation of attention that goes beyond heuristic gating.
2. Unifying lens on gated models. Table 1 methodically maps several models into the same objective, and shows Mamba2 as a limiting case. This is an interesting contribution.

**Weaknesses:**

1. Scaling story is mixed. At 760M, Palimpsa underperforms Gated DeltaNet on the averaged suite, weakening the claim that metaplastic updates are broadly advantageous; an analysis isolating why performance flips with scale is needed.
2. Fairness of comparisons needs tightening. Table 5 shows Palimpsa uses more layers yet half state size to keep state budgets comparable. Please provide compute‑matched and parameter‑matched comparisons (same layers/expansions).
3. Limited long‑context evaluation. Claims about a tunable memory horizon would be more convincing with established long‑range tasks.
4. There is a lack of discussion on TTT/Titans models. The emperical baselines are borrowed from Titans, so why these baselines are missing?

**Questions:**

See weaknesses,

---

> ### Author Response · Authors · 2025-12-03
> **Rebuttal**
>
> Thank you for your encouraging review. We are pleased you found our unifying lens on gated models interesting.
>
> 1. **Scaling at 760M:** We hypothesize that the inductive bias of having a metaplastic attention matters less as the model size increases. We are investigating the cause of this as scale increases.
>
> 2. **Fairness of Comparisons:** Halving the state budget ($d_k$) meant Palimpsa used *fewer* parameters compared to Gated DeltaNet. To ensure a fair comparison, we increased the number of layers to match the total parameter count. While this results in slightly more states, the layer ratio remains comparable (21 vs 19 in the most extreme case), ensuring the models are matched on total capacity. We believe that this is a more meaningful comparison than matching compute, since in such workloads, the memory bandwidth (HBM) is generally the bottleneck. [1]
>
> 3. **Titans/TTT Baselines:**  We agree that a comparison would have been interesting. However, reproducible code for these baselines are not provided by the authors.
>
> ### References
>
> [1] Yang, S., Kautz, J., & Hatamizadeh, A. **Gated Delta Networks: Improving Mamba2 with Delta Rule.**

---

### Official Review · Reviewer_etLB · 2025-11-01

**Soundness:** 2
**Presentation:** 2
**Contribution:** 3
**Rating:** 2
**Confidence:** 3

**Summary:**

The paper proposes what could be considered an extension of LongHorn or MesaNet, adding a Bayesian framework inspired by Bonnet et al. that explicitly allows forgetting, in an input-dependent way. It proposes a new token mixer in state space models, addressing in-context catastrophic forgetting and catastrophic remembering.

**Strengths:**

1. It's interesting to see a Bayesian scheme for meta-plasticity, combining ideas from LongHorn and Mesanet.
2. The authors attempt to structure the field a bit, e.g. by showing methods like MAMBA2 are a special case of their work. Table 1 is interesting in that respect, but could be made clearer.
3. They achieve competitive results, especially for 'smaller' models.

**Weaknesses:**

1. The paper needs significant restructuring and rewriting, in my opinion.
- Half of the main paper is spent on contextualization and related work. They start from a too broad context of self-attention and in-context learning, while actually their work applies to state space models. Stating that explicitly from the start, would have made the actual contributions a lot clearer.
- A lot of the relevant content of the paper, related to the actual method, is moved to the supplementary material.
- Supplementary material is not well structured. It feels like the authors abuse the supplementary material to bypass the page limit. Text in the main paper is too verbose and not focusing on the essence. At the same time, technical descriptions in the main paper are vague and handwavy, with important details left out. Instead of pointing to a specific section in the appendix where those details can be found, it's left to the reader to go dig in the supplemental material to try and find them.
This makes it hard for the reader to really grasp the technical details.

2. Poor analysis of the proposed method
The experimental results are limited to 2 tables only, basically comparing the method as a whole against competing methods, once on the MAD benchmark and once on the fineweb-edu dataset.
- There is no ablation study.
- There is no sensitivity analysis on hyperparameters.
- There are no further analyses trying to understand the behavior of the model.
In particular, I would be interested to know when the model decides to forget. Even if only anecdotal, some examples or visualizations could give some insights. In that context, I've always found it weird the forgetting and input gates depend on the input, not on a combination of input and state.
Also interesting would be some insights in the distribution of the gate values.

3. Difficult to reproduce.
I would have a really hard time reproducing the method, based on the information provided (even including the supplementary materials). There's no mention in the paper about the authors planning to make code or pretrained models publicly available.

**Questions:**

1. The paper states some explicit differences of the proposed method compared to MesaNet (end of related work section). Which of these differences are most important to improve the results ? Is it the Bayesian framework per se ? Or the non-scalar beta ? Or the linking of the input gate to the forgetting gate ?

---

> ### Author Response · Authors · 2025-12-03
> **Rebuttal**
>
> Thank you for your encouraging review. We are pleased you found our Bayesian scheme for metaplasticity in attention-based models interesting.
>
> 1. **Context and Scope:** We acknowledged your comment; in future revisions, we will lighten the contextualization. However, we believe it remains essential to place this work in the context of self-attention and in-context learning. This entire work is based on the idea of treating in-context learning as a continual learning problem. While our work connects naturally to SSMs because it replaces the KV cache with a fixed-size state, it does not derive from SSM theory but rather from (gated) linear attention. Input-dependent gating is crucial for selectivity; it allows the model to filter out irrelevant inputs, preventing them from overwriting the state.
>
> 2. **Reproducibility:** We already mentioned the code will be made available in the text. Furthermore the supplementary materials already provided the necessary kernel files. The github repository link will be made public after publication. This work is built within the framework of the Flash Linear Attention (FLA) GitHub repository.
>
>
> 3. **Differences from MesaNet:** The main difference is the non-scalar $\beta$. This is essential to yield the phenomenon of metaplasticity, where each single state has a learnable learning rate. The Bayesian framework *per se* did not provide a performance advantage in this case, as both interpretations yield the same math. However, the Bayesian framework provides a mathematical foundation for uncertainty estimates, which will be used in future work.

---

### Note · Authors · 2025-12-03

**Comment:**

We have decided to withdraw the submission to allow sufficient time to fully conduct the experiments requested by the reviewers. We believe incorporating these results is essential to meet the paper's claims and standards.

**Withdrawal Confirmation:**

I have read and agree with the venue's withdrawal policy on behalf of myself and my co-authors.